# Scale-Up of Doppler to Improve Intrapartum Fetal Heart Rate Monitoring in Tanzania: A Qualitative Assessment of National and Regional/District Level Implementation Factors

**DOI:** 10.3390/ijerph17061931

**Published:** 2020-03-16

**Authors:** Marya Plotkin, John George, Felix Bundala, Gaudiosa Tibaijuka, Lusekelo Njonge, Ruth Lemwayi, Mary Drake, Dunstan Bishanga, Barbara Rawlins, Rohit Ramaswamy, Kavita Singh, Stephanie Wheeler

**Affiliations:** 1USAID’s Maternal and Child Survival Program/Jhpiego, Washington, DC 20036, USA; Barbara.rawlins@jhpiego.org; 2UNICEF, Dar es Salaam, Tanzania; jgeorge@unicef.org; 3Ministry of Health, Child Development, Gender, the Elderly and Children, Dar es Salaam, Tanzania; bundala.felixambrose@yahoo.com; 4USAID’s Maternal and Child Survival Program/Jhpiego, Dar es Salaam, Tanzania; Gaudiosa.tibaijuka@jhpiego.org (G.T.); Lusekelo.njonge@jhpiego.org (L.N.); Ruth.lemwayi@jhpiego.org (R.L.); Mary.drake@jhpiego.org (M.D.); dbishanga@gmail.com (D.B.); 5Department of Maternal and Child Health, Gillings School of Global Public Health, University of North Carolina at Chapel Hill, Chapel Hill, NC 27599, USA; ramaswam@email.unc.edu (R.R.); singhk@email.unc.edu (K.S.); 6Department of Health Policy and Management, Gillings School of Global Public Health, University of North Carolina at Chapel Hill, Chapel Hill, NC 27599, USA; Stephanie_Wheeler@unc.edu

**Keywords:** Doppler, scale-up, intrapartum care, fetal heart monitoring, Tanzania, qualitative

## Abstract

High-quality intrapartum care, including intermittent monitoring of fetal heart rates (FHR) to detect and manage abnormalities, is recommended by WHO and the Government of Tanzania (GoT) and creates potential to save newborn lives in Tanzania. Handheld Doppler devices have been investigated in several low-resource countries as an alternative to Pinard stethoscope and are more sensitive to detecting accelerations and decelerations of the fetal heart as compared to Pinard. This study assessed perspectives of high-level Tanzanian policymakers on facilitators and barriers to scaling up use of the hand-held Doppler for assessing FHR during labor and delivery. From November 2018–August 2019, nine high-level policymakers and subject matter experts were interviewed using a semi-structured questionnaire, with theoretical domains drawn from Proctor’s implementation outcomes framework. Interviewees largely saw use of Doppler to improve intrapartum FHR monitoring as aligning with national priorities, though they noted competing demands for resources. They felt that GoT should fund Doppler, but prioritization and budgeting should be driven from district level. Recommended ways forward included learning from scale up of Helping Babies Breathe rollout, making training approaches effective, using clinical mentoring, and establishing systematic monitoring of outcomes. To be most effective, introduction of Doppler must be concurrent with improving case management practices for abnormal intrapartum FHR. WHO’s guidance on scale-up, as well as implementation science frameworks, should be considered to guide implementation and evaluation.

## 1. Introduction

Globally, an estimated 2.1 million early neonatal deaths and 2.6 million stillbirths occurred in 2015, including 1.3 million intrapartum stillbirths [1]. Almost all of these (98%) occurred in low- and middle-income countries (LMICs) [1,2,3]. Skilled birth attendance, where intrapartum and newborn care comply with globally recommended quality standards, could prevent a substantial portion of these deaths [4]. 

Abnormal fetal heart rate (FHR) in the intrapartum period can indicate a hypoxic state in a fetus resulting from interruption of placental blood flow [5]. Because abnormal FHR is a potential predicator of newborn asphyxia, FHR monitoring is important for quality intrapartum care. Conversely, poor quality intrapartum FHR monitoring contributes to intrapartum stillbirths [6]. In Tanzania, studies have provided strong evidence of fetal heart abnormalities as a predictor of fresh stillbirth, birth asphyxia, and newborn death [7]. Improvements to intrapartum monitoring have had demonstrated results, as in a 1989 study in southwest Tanzania where an intervention related to intrapartum monitoring was associated with a reduction of perinatal mortality from 71 to 39 deaths per 1000 births [8]. Despite this evidence, quality of intrapartum FHR monitoring, both upon admission to labor and delivery services and intermittently throughout labor, is often poor in Tanzania [6].

The World Health Organization (WHO) recommends intermittent FHR monitoring during labor in the LMIC setting, but does not endorse a particular tool [9]. In LMIC health facilities, the Pinard stethoscope is widely used to assess FHR in the intrapartum period, rather than cardiotocography (the standard of care in high resource countries) or handheld Doppler device [10]. However, multiple randomized controlled trials in sub-Saharan Africa have demonstrated that Doppler is superior to Pinard stethoscope at detecting abnormal FHR [11,12,13,14,15]. And although evidence on client preference for Doppler over Pinard is not strong, a small study in South Africa showed that laboring women preferred Doppler over Pinard for FHR monitoring [16]; and in a qualitative study in Tanzania, women in labor showed a strong liking for Doppler for continuous FHR monitoring [17]. Although further evidence is needed, a growing base of findings suggests that using Doppler to monitor FHR during labor in the LMIC facility setting may improve both quality of clinical care and women’s experience of intrapartum care. 

In 2008, following a call for a single plan to address reproductive, maternal, newborn, and child health strategies, the Tanzanian Ministry of Health (now the Ministry of Health, Community Development, Gender, the Elderly and Children) developed “The National Road Map Strategic Plan to Accelerate Reduction of Maternal, Newborn, and Child Deaths in Tanzania 2008–2015,” referred to as the One Plan [18]. The One Plan included basic emergency obstetric and newborn care (BEmONC) targets, such as intermittent FHR monitoring through use of the partograph. The Sharpened One Plan, created in 2013, emphasized obstetrics and family planning and focused on the Western and Lake Zones of Tanzania [19]. An evaluation of the Sharpened One Plan found that achievements fell far short of targets; for example, the proportion of public health facilities offering BEmONC services was 45% in 2015 against a target of 70% [19]. Currently, the policy framework for improving maternal and newborn care is governed by the One Plan II, the country’s second national strategic plan, which covers 2016–2020. 

Tanzania’s National Service Quality Improvement Tool of 2013 recommends FHR monitoring every 5 minutes when a woman is in the second stage of labor [20]. Similar to WHO guidance, the tool says nothing about whether Doppler or Pinard stethoscope should be used to monitor FHR.

The Helping Babies Breathe (HBB) training approach and set of materials are designed to boost health care providers’ newborn resuscitation skills in the low-resource health facility setting. HBB has been used in many LIMCs to improve the quality of immediate newborn care [21]. As an intrapartum capacity-building intervention, HBB sets a precedent for scale-up that intrapartum FHR monitoring can use as a model. The Tanzanian national HBB program, which started in 2013, was scaled up to 15 of Tanzania’s (then) 25 regions with a 1-day, onsite training for health care providers, provision of supplies and equipment, and post-training mentoring to health care providers. HBB rollout has been evaluated in Tanzania in terms of programmatic approach to scale-up [21], validation of tools and approaches for training [22,23], and cost-effectiveness [24,25]. 

This analysis presents views from high-level Tanzanian policymakers and subject matter experts (SMEs) about the potential use of Doppler for FHR monitoring in health facilities at large scale. The interviews centered on the following research questions: What are the facilitators and barriers to scale-up of Doppler?What lessons can be learned from Tanzania’s experience of scale-up of the HBB program?Who needs to do what to scale up Doppler in Tanzania?

The study will be useful to policymakers and program implementers developing policies, protocols, guidelines, and standards to improve intrapartum care in Tanzania and similar settings. 

## 2. Methods

This qualitative study used in-depth interviews to elicit views and opinions of policymakers and SMEs from the maternal and newborn health fields on the environment and precedents for scale-up of Doppler for FHR monitoring during intrapartum care in Tanzania. The analysis uses a social ecological framework to contextualize findings.

### 2.1. Social Ecological Framework for Scaling Up Doppler in Intrapartum Care

The social ecological model has proven useful for exploring potential barriers and facilitators to health service use [26]. The current study uses this model to examine national, subnational, and organizational factors associated with the research question, What are the barriers and facilitators to scale-up of Doppler in Tanzania? Figure 1 presents the adapted social ecological model, with the health facility at the individual level. Within the layers of environmental factors in Figure 1, actions or conditions associated with “what needs to be done” have been described. The current study does not answer questions associated with sub-n-national, community or health facility levels, rather the focus is the two outermost levels (policy and organizational levels).

We used a stakeholder analysis framework to examine the research question, Who needs to do what to scale up Doppler in Tanzania? To assign roles to each key stakeholder in the framework, we took suggestions made by interviewees or deduced by the study team and plotted using a diagram. 

### 2.2. Participants

A total of nine interviewees were selected. All were Tanzanian. The number of interviews was not predetermined; rather, we designed the interviews to balance input from different branches of the GoT and agencies. Interviewees included high-level Government of Tanzania (GoT) policymakers and national maternal and newborn SMEs at professional organizations, universities, and donor agencies. Regional health authorities from Mara region of Tanzania were also included. Mara is part of an ongoing study examining perinatal mortality rates in facilities using Doppler. Mara region is supported by the Maternal and Child Survival Program (MCSP), a global 5-year cooperative agreement funded by the United States Agency for International Development (USAID) to introduce and support scale-up of high-impact health interventions. Table 1 lists the interviewees’ positions and organizations.

### 2.3. Data Collection Procedures and Study Tool

We selected three themes from Proctor’s classification of implementation outcomes [27] to correspond to the early stage of implementation of Doppler in Tanzania. The selected themes (costs, appropriateness, and acceptability) are a subset of a larger group of implementation outcomes (acceptability, adoption, appropriateness, costs, feasibility, fidelity, penetration, and sustainability). These themes led to specific questions and prompts in the questionnaire (Table 2). 

Interviews took place between November 2018 and August 2019. Members of the research team conducted the interviews using the standardized interview guide (available in the Appendix A) after obtaining interviewees’ written consent to participate in the study. Interviews were audio recorded and interviewers took notes, which were cross-referenced during the analysis. Interviews were conducted in English (n = 8) or Swahili (n = 1), depending on the interviewee’s preference. Transcripts were produced from audio recordings for analysis, and translated into English when needed. 

### 2.4. Data Analysis

Because of the small number of interviews, analysts (Marya Plotkin, John George) used manual coding to identify themes and codes. We created a codebook delineating themes drawn from the feasibility domains (Table 2), and used color schemes to highlight and annotate text and relate themes and codes in the transcripts. The coding process followed the grounded theory tradition [28], allowing additional themes and codes to arise during analysis. After the initial pass through the transcripts, the two analysts agreed upon a consolidated list of codes. Using the expanded group of codes, they reviewed the transcripts again, and created a summary of findings and quotes by theme and code. 

## 3. Results

### 3.1. Costs

#### 3.1.1. Purchasing Dopplers for Health Facilities

Although several interviewees felt that the local government (through facility and district funds) and the central government and implementing partners (Tanzanian and international civil society organizations) should share responsibility for purchasing Dopplers, most stressed that Dopplers should be purchased with district or facility funds. The specific mechanisms identified were direct health facility financing, internally generated revenue from health insurance schemes (Interviewee 3), government funds (in the form of district-allocated health budgets) (all participants), nongovernmental organizations (NGOs), and private companies, such as in a corporate social responsibility initiative in Geita where companies donated ultrasound machines (Table 3).

To work around this, one participant stressed the importance of priority building so that facilities can plan, budget, and procure Dopplers, indicating that without any one of these steps the device would not reach facilities. 

Several participants mentioned including Doppler in facility budgets as a necessary step: 


*If there is anything that is not in the budget, that would be a barrier for the facility to buy. It doesn’t matter how much it costs, it’s the issue of entering it into the normal yearly budget of every facility.*


One participant raised the cost of a Doppler compared to a Pinard, along with other factors that might eventually be evaluated as part of a cost/benefit analysis: 


*For example, a Pinard stethoscope costs about 5 US dollars compared to the Doppler, which costs about 200 US dollars. If you compare in terms of the prices, the Doppler looks to be expensive. But in terms of functionalities, we find that Doppler saves a lot of time, and it’s easier to use.*


#### 3.1.2. Procurement Processes

Although all participants felt that Dopplers were affordable and within the means of health facility budgets, several raised concerns about delays or roadblocks associated with facilities procuring equipment. One participant cited an example of a health facility that did not have a refrigerator or consistent power but could not navigate the system to procure one:


*I proposed, why don’t you buy a fridge powered by solar? They said they can do that, but after 6 months it was still on procurement. That year the budget closed with 6 million shillings still in the budget (and no refrigerator).*


### 3.2. Appropriateness

#### 3.2.1. Training 

Several interviewees stressed the importance of workplace-based training on the use of Doppler and the need for subsequent mentoring:


*Updates should focus on how to give high-quality care, on an on-job training basis.*



*The training should be mentorship, not classroom learning… [W]e can now see vividly that this helps someone to acquire that skill.*


Two interviewees recommended a low-dose, high-frequency (LDHF) approach to training (Interviewees 7, 8). One interviewee thought that adding a day to existing BEmONC in-service training would be the best way to scale up the program nationally, unless an external donor was willing to fund a HBB-like standalone training. Additionally, clinical management following detection of abnormal FHR was described as critical:


*The issue is not just to put the Doppler at the bed of a pregnant woman, but also to know the concept behind the Doppler. Otherwise it can be there, but the interpretation of that data? The midwife may not think that the woman is in danger and take action.*


Two interviewees stressed the need to incorporate facility-level competence into existing supervision systems: 


*Use the Regional and Council Health Management Teams (RHMT and CHMT) to train the providers at the level of the facility. I think that will really cut the cost, because we know that the CHMT and RHMT have regular supportive supervision at the facility … while they are going for that regular supervision, they go with the plan of training people.*



*We as regional supervisors, progressively need to supervise and sensitize … [W]e must make sure we work together with Councils as they have the capacity to reach all facilities. Emphasis must go on the facility in-charges who ensure that intended women get the service.*


One interviewee stressed the need to develop or incorporate Doppler into clinical guidelines:


*I think first and foremost, we need to develop guidelines—guidelines for use of Doppler, specifically.*


Two participants mentioned a lack of available evidence on cost and cost-effectiveness needed for the government to make a decision on Doppler scale-up. 

#### 3.2.2. Levels of Care

One participant was concerned with whether Doppler would be equally useful at lower and higher level facilities, pointing out that many regional and some district hospitals already use Doppler. 

#### 3.2.3. Practicability Barriers

Multiple interviewees mentioned lack of a reliable power source at health facilities to recharge the battery for the hand-held device as a potential barrier to scale-up. Recommendations included looking into manual or solar-powered Doppler devices to accommodate this challenge. One participant noted the need for integrated solutions to improve quality of intrapartum care: 


*The other note I want to make is that Doppler alone is not going to improve intrapartum care…. [A]reas like training of the personnel and staffing should come together.*


#### 3.2.4. Lessons Learned from HBB Scale-Up

Multiple respondents noted the link between HBB as an approach to improve newborn outcomes and use of Doppler for FHR monitoring. Others recognized that HBB scale-up significantly improved newborn care in Tanzania: 


*HBB has contributed a lot to improve the newborn’s conditions … [I]t has rescued a lot of babies who had nearly died but were saved due to availability of these services in health facilities.*



*HBB is among the very successful programs that has been taken up here in Tanzania.*


One participant was particularly concerned about availability of equipment (bag and mask in newborn size) as a barrier to HBB scale-up: 


*The barrier, or can I say challenge, to HBB was the availability of equipment … [I]n most of the government facilities, you find that there are so many clients … even when the equipment is there to help the baby survive, it is inadequate in quantity.*


This concern may be of note in looking at needs for assessing Doppler quantities available for provision of care. One concern raised about HBB scale-up in Tanzania was a perception that HBB (implementation, programmatic outcomes) has not been monitored closely. As one interviewee described:


*HBB in most parts of this country is not monitored—even at the regional level. It has been scaled up and nobody is concerned with monitoring—this is bad. You should monitor everything to see its effectiveness, see its applicability, to see the challenges.*


### 3.3. Acceptability and Alignment with National Priorities

At least one participant stressed the link between the current national strategy (One Plan II) and improving intrapartum care. The participant noted that the emphasis on delivering babies in the health facility setting had not translated to improved perinatal mortality: 


*Over the past 15 years … we have seen a lot of emphasis on getting women to deliver at the health facilities. And we’ve done very well on that. We’ve moved from less than 40% hospital delivery to almost 63%, but we have not seen a significant reduction in infant mortality. And we ask why? Because it looks like we have moved these deaths from the community to the health facility … [T]hese facilities are not prepared to receive that influx of mothers. You see? So currently the government putting a lot of emphasis on improving the quality of intrapartum care.*


The same participant was highly positive about use of the Doppler in Tanzanian health facilities for improved FHR monitoring, arising from recent studies in Tanzania:


*Our data supports that if you use Doppler… any Doppler, the quality of intrapartum care becomes better and the outcomes are better. I think that’s the most important driver because we all want to reduce perinatal mortality… So, if somebody’s asking you, why do you want this? We have the evidence.*


#### 3.3.1. Acceptability to Health Care Providers

Almost all interviewees viewed Doppler positively in terms of how they believed health care providers would like Doppler and its utility in intrapartum care. Several interviewees were midwives and saw the benefit to clinical management (Interviewees 3, 4, 6). One interviewee was certain that Doppler would make birth attendants’ jobs in crowded health facilities easier and lead to better care for women:


*If we have the room to get a device which can assist the midwife to monitor fetal heart rate, I think we should be very positive. We are always telling our people to shout for help, and if this device really assists them to shout for help, why should we wait? We need it right now. I’m not the one budgeting, but as a midwife, I think we need this device. We need to advocate to the management to see the importance of this one.*


One interviewee cautioned against expecting an enthusiastic reception of Doppler for FHR monitoring, citing experience from a recent study in Tanzania in which midwives did not prefer Doppler over the Pinard stethoscope; rather, they preferred the device they were most familiar with (Interviewee 8, Aga Khan University). 

Another interviewee stressed the need for both health care providers and women entering maternity services to buy in to use of Doppler for FHR monitoring. (Doppler makes the fetal heartbeat audible to both the health care provider and the woman in labor.)


*One of the key issues which needs to change is attitude (of the health care provider). Change your attitude, and then we will make sure that the new technology will improve intrapartum care … and not only the health care provider, mothers too. Because some of them don’t like to hear the fetal heartbeat. We need to get the mother and health care providers aware the advantage of using the Doppler.*


Another mentioned that Doppler would simplify FHR monitoring for health care providers, who currently use their watches to time heartbeats when using a Pinard stethoscope: 


*Availability of Doppler actually simplifies work for the provider in reducing the need for counting and using a watch (associated with use of Pinard fetoscope).*


However, one interviewee cautioned against being overly optimistic about Doppler’s acceptability among health care providers, citing Rogers’ Diffusion of Innovation theory [29]: 


*Any innovation which comes in, you don’t expect to be accepted right away.… [B]ecause people want to stay in their comfort zone. So, if a person is used to having a Pinard stethoscope, it’s maybe comfortable to continue using the Pinard stethoscope because he has been doing that over years and years.… So, you have those kinds of people who will take the innovation when they get it, but you have those people who will not take it from the beginning.… We should be aware of that. That even though this innovation may be good, we should not expect immediate acceptance by all.*


#### 3.3.2. Alignment with National Priorities

Overall, there was general agreement that Doppler aligned with national priorities:


*When we read the Road Map you see that it aims at certain goals, one of them being the reduction of maternal death as well as newborn and perinatal death. So, proper monitoring during labor will definitely help to identify babies in distress and respective intervention can be performed, timely.*


However, participants’ listing of priorities for improving intrapartum care varied in terms of how closely Doppler aligned with perceived national priorities. As Table 3 shows, some respondents mentioned bigger picture issues, such as human resource shortage/increasing the number of skilled birth attendants, and provision on the government’s ability to provide comprehensive emergency obstetric and newborn care (CEmONC).

Three respondents linked use of Doppler with the One Plan II, and one stated the importance of describing Doppler in national plans and clinical guidance:


*I think the best way (to scale up Doppler) is to make sure that the intervention is clearly shown in the One Plan II. From the One Plan II, you can get all of these interventions focusing on the Doppler, where we can plan to train all of these providers, make sure the Dopplers are available, [and] the distribution is good.*


Although not all interviewees specifically noted FHR monitoring as a high priority for intrapartum care, they all acknowledged that using Doppler to improve FHR monitoring aligns with government priorities. One respondent tied use of Doppler for FHR monitoring to the importance of reducing newborn deaths in health facilities:


*One of the issues, which is alarming, is the number of newborns dying in our health facilities. If you go through our statistics, the number of newborns reported dead, one of the leading causes is birth asphyxia … so if you are in a position to establish that at the time the woman is delivering, that baby survives, that will be good!*


Another participant mentioned that to link Doppler with national priorities outlined in the One Plan II, data will need to be made available to show the outcomes of using Doppler for FHR monitoring:


*We need to keep the data to see how this will help our people so that the government has a clear reason and a clear vision for using it to reduce death.*


### 3.4. Stakeholders’ Roles and Responsibilities

Respondents mentioned roles and responsibilities consistent with the outer two levels of the social ecological framework (policy level and organizational/nonstate actor level), and commented on the roles of the government (national, regional, and district level), funding agencies, NGOs, and professional associations (Figure 2). 


*Key: Ministry of Health, Child Development, Gender, the Elderly and Children: MOHCDGEC; President’s Office/Regional and Local Governance: PO-RALG; Tanzania Midwifery Association: TAMA; Tanzania Nurse and Midwives Council: TNMC; Association of Gynecology and Obstetrics of Tanzania: AGOTA*


Interviewees described the role of the GoT (MOHCDGEC and PO-RALG) as setting national standards (in line with national priorities elucidated in the One Plan II) (Interviewee 5, PO-RALG) and synthesizing current evidence on Doppler to make informed decisions. 

At the subnational level, virtually all interviewees mentioned the need for district councils and health facilities to include Doppler in their budgets. 

One interviewee stressed that implementing partners/NGOs could best help with capacity-building (training, supervision, quality control) for health care providers: 


*There are so many updates sometimes, it is not easy for the government facilities to get it … the implementing partners have updates and can build capacity of the health care providers.*


Several interviewees mentioned that professional associations have a role to play in scale-up of Doppler, “since it is the role of the professional organization to help the government.” This included capacity-building of health care professionals. Another respondent mentioned that the Tanzanian Midwifery Association (TAMA) could play an important role in advocating for nurses and midwives (who constitute the majority of skilled birth attendants in Tanzania) to use Doppler, stating, “If TAMA advocates, health care providers will listen.” 

One respondent cautioned about the need for the GoT to take a lead role, particularly in financing scale-up of Doppler:


*As far as I know, the Government of Tanzania has not really budgeted, or adopted the use of the Moyo (Doppler) throughout the country, hopefully they would be able to do that. Without that, a scale-up of Moyo (Doppler) will suffer from the same disease: donor dependence and lack of buy-in from the local resources.*


## 4. Discussion

This qualitative assessment of views of high-level policymakers and SMEs found that most were optimistic about Doppler’s potential to improve intrapartum FHR monitoring in Tanzania and its potential scale-up. Respondents provided specific feedback on what might be needed at the GoT/international donor level (guidance on clinical management and supervision and funding of training and Dopplers); and at the nonstate actor level (guidance for training, supervision, and quality assurance and assistance in using monitoring system). Table 4 summarizes the study findings relevant to national- and organizational-level factors. 

Lessons from the scale-up of HBB include the need for a monitoring system as well as effective training and follow-up methodologies. 

Respondents saw links to national priorities laid out in the One Plan II, as well as the potential for competing priorities for improving intrapartum care. Several interviewees noted that the ultimate goal was not scale-up of Doppler, but improved FHR monitoring and action to prevent perinatal mortality. This echoes perspectives on scale-up of kangaroo mother care (KMC), where some experts describe simply setting aside a room for KMC rather than change case management for newborns [30]. To avoid “empty” scale-up of Doppler, GoT policymakers may consider substantive ways to support good case management practices, such as tools and clinical mentoring for health care providers. The UK’s National Institute for Health and Care Excellence job aide “Fetal Monitoring during Labour” [31] is an excellent resource.

Several studies have compared Doppler’s efficacy to Pinard in detecting FHR abnormalities [11,12,13,14,32]. These studies inform policy through evidence, but provide little insight on district-, regional-, and facility-level scale-up. A few publications shed light on the wider context for scale-up. For example, one study in Tanzania found that midwives who used both Doppler and Pinard to monitor FHR in a randomized controlled trial in a rural hospital tended to prefer the device with which they were most familiar. Recommendations included incorporating the newer Doppler technology into both pre-service and in-service education [33]. In another study in Kagera, Tanzania, authors noted that incorporating Doppler into the admissions workflow was feasible and did not greatly add to the time needed to admit women to labor and delivery services [34]. While these two studies present contextual findings at the health facility level to help policymakers address needs and rationale for scale-up, the evidence base is notably limited. Further studies or program learning that contextualize factors for successful scale-up of Doppler are needed, at all levels. 

A number of useful resources are available to assist GoT policymakers develop better guidelines for case management of FHR abnormalities, including recent WHO guidance on synthesizing national policy using systematic reviews or program evaluations [35] and on taking interventions to scale [36]. If the GoT decides to scale up Doppler, a scale-up framework that includes planning, costing, financing, implementation, and monitoring will be important to both facilitate engagement and serve as a coordinating mechanism. Several frameworks provide guidance on monitoring implementation, including Proctor and colleagues’ implementation outcomes framework, which ties outcomes to stage of implementation [27]. Additionally, teams have used the consolidated framework on implementation research (CFIR) [37] to support health service improvements, including a study on improving hospital service quality in Kenya [38].

Interviewee’s recommendations on improved training methodologies, such as low dose, high frequency (LDHF) and onsite training, are supported by the literature. A systematic review of in-service training showed that targeted, repeated interventions lead to better training outcomes [39]. A recent study in Mozambique found that LDHF training on newborn resuscitation helped improve midwife performance [40], and in Tanzania a supportive intervention to HBB resulted in higher retention of newborn resuscitation skills [23]. A cluster randomized trial in Ghana found that in health facilities using an onsite, LDHF approach to training on intrapartum care, the relative risk of newborn mortality and intrapartum stillbirth significantly declined up to a year following training, leading the authors to recommend use of this training approach [41]. 

Publications on taking HBB to scale in Tanzania reinforce the correlations made by participants between HBB scale-up and Doppler scale-up. The length of training for HBB was 1 day [22]; similarly, several research studies have reported 1-day training on Doppler [6,19,42]. Participants in the current study stressed the need for clinical mentorship and follow-up to retain health care providers’ skills. This was noted in HBB scale-up as well, as 87% of health care providers passed a competency exam immediately following training, but only 56% passed 4–6 months following training. In Tanzania, this drop-off in skills caused a programmatic shift to how health care providers were trained and supported following training [43]. The original 1-day HBB training was expanded by a half-day following development and implementation of a structured on-the-job training guide to standardize LDHF practice, review of service delivery data, and mentorship on clinical practice and data recording [23]. Scale-up of Doppler would do well to incorporate these lessons from HBB scale-up, if it moves into a scaled-up training phase. 

Policy and clinical guidance for improved intrapartum care is not the only important element of scale-up: as several respondents noted, government and donor funding must also support scale-up of Doppler. Although newborn deaths account for 39% of deaths of children under age 5 years in Tanzania [43], only a tiny proportion of funding is allocated to address this issue. In 2010, for example, $1.5 million was allocated to programs benefiting neonates, whereas in the same year, $208 million was allocated to reproductive health (family planning and sexually transmitted infections, including HIV) [18]. An assessment estimated that national scale-up of HBB training and equipment in Tanzania would cost roughly $4 million (range $2.9–$4.3 million), an average of $600 per health facility [25]. Similarly, studies have evaluated the effectiveness of the LDHF training approach. In Ghana, for example, a study found that LDHF training for improved intrapartum care had a cost-effectiveness of $53 per disability-adjusted life year (DALY) averted, using intrapartum stillbirth and newborn death as the measure from which DALY was calculated [43]. Cost and cost-effectiveness analyses of use of Doppler for intrapartum FHR monitoring, currently missing from the literature, would be useful to the GoT and other governments as they consider scale-up of Doppler. To fill this gap, more studies and well-documented program learning that look at operational aspects of incorporating Doppler into health services are needed.

### Limitations

This study interviewed a small number of respondents. However, not many people are working at the national policymaking level on intrapartum care provision as provided by the GoT health services. Additionally, we noted a high level of agreement/saturation in the responses, which leads us to believe that the number of respondents may have been sufficient to reach saturation.

This analysis of use of Doppler for FHR monitoring did not provide discussion around types of Doppler best suited for Tanzania’s needs. We acknowledge that there are multiple types of Doppler devices and further investigation will be needed into which devices are most suitable for the Tanzanian health care setting. 

Although this study analyzed environmental factors associated with scale-up of Doppler for FHR monitoring in Tanzania, community-, facility-, and individual-level factors are equally important. More studies are needed in these areas.

## 5. Conclusions

Interviewed experts and policymakers consider scale-up of Doppler for improving intrapartum FHR monitoring as aligning with national health priorities in Tanzania. Linkages and lessons learned from the scale-up of the national HBB program include implementing a structured LDHF training approach that includes clinical mentoring and monitoring of results. It will be useful for the GoT to assess evidence on benefits of Doppler for FHR monitoring using WHO guidance for synthesis of policy, and if Doppler is adopted at scale, to create a scale-up framework from early phases to enhance coordination. If scale-up is pursued, sufficient resources, training and infrastructure to support effective use of Doppler for FHR monitoring and intrapartum case management must be in place to overcome barriers described by participants. 

## Figures and Tables

**Figure 1 ijerph-17-01931-f001:**
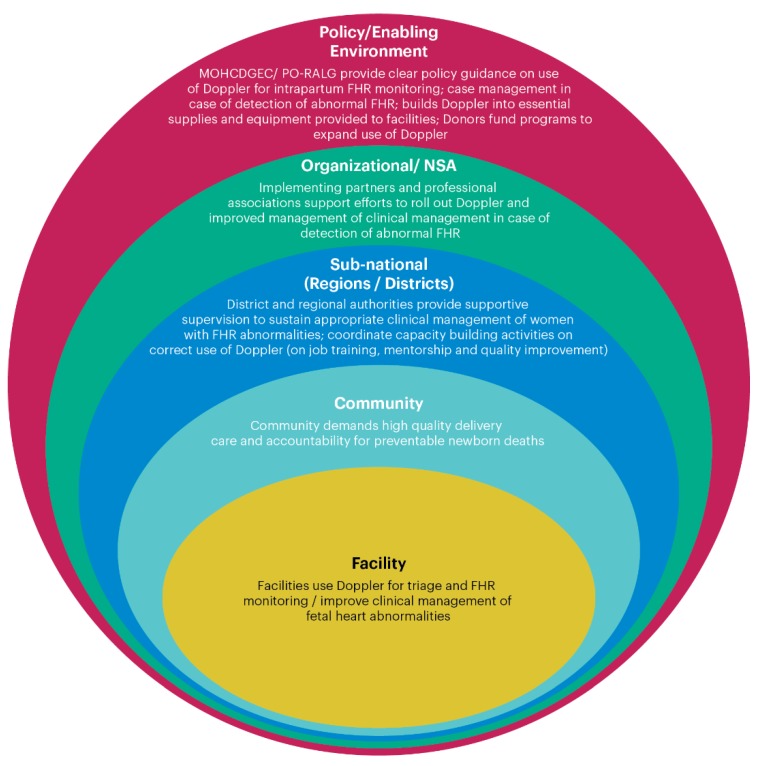
Social ecological framework: needs associated with every environmental level for scale-up of Doppler. NSA: Non-state actors.

**Figure 2 ijerph-17-01931-f002:**
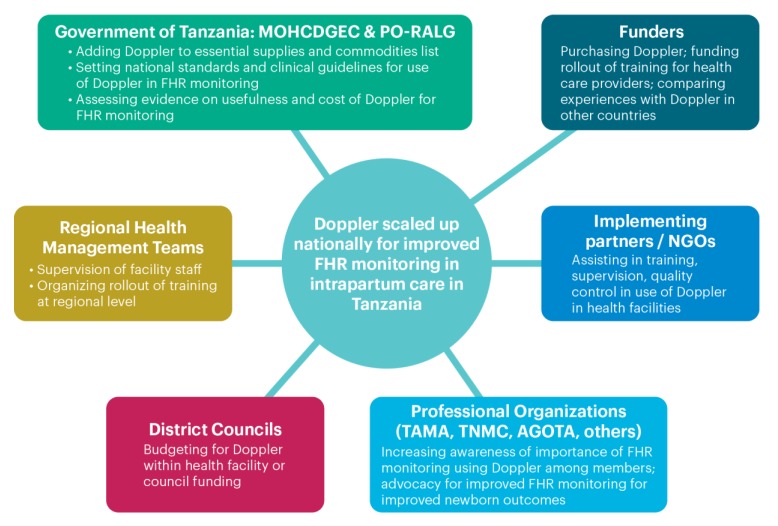
Roles and responsibilities of at policy, organizational, and subnational level for scale-up of Doppler.

**Table 1 ijerph-17-01931-t001:** Study participants.

Number of Interviewees and Roles	Organization
(2) Senior advisors	Reproductive Maternal, Newborn, Child, and Adolescent Health Department, the President’s Office, Regional and Local Government (PO-RALG)
(2) Senior Advisors	Reproductive and Child Health Section, Ministry of Health, Community Development, Gender, the Elderly, and Children (MOHCDGEC)
(1) Registrar	Tanzania Nursing and Midwifery Council
(1) Reproductive and child health services coordinator	Regional Health Management Team, Mara
(1) Research scientist, subject matter expert	Haydom Lutheran Hospital
(1) Subject matter expert	Aga Khan Medical University
(1) Program specialist, Maternal, Newborn and Child Health	U.S. Agency for International Development, Tanzania

**Table 2 ijerph-17-01931-t002:** Domains for scale-up of Doppler for intermittent fetal heart rate monitoring in Tanzania.

Domain	Questions
Costs	What is the current resource availability for improving intrapartum care? What financial and other resources were needed from external donors for rollout of the Helping Babies Breathe initiative from the Ministry of Health, Community Development, Gender, the Elderly, and Children (MOHCDGEC)? What affordability barriers do you anticipate if Doppler were to be scaled up to all facilities providing maternity services? What facilitators do you anticipate?
Appropriateness	What are current priorities for improving intrapartum care in public health facilities in Tanzania? How does Doppler align with national priorities? What resources would be needed to scale up use of Doppler in intrapartum care in Tanzania? from the MOHCDGEC? from nongovernmental organizations? from external donors? What human resources and systems barriers do you anticipate if Doppler were to be scaled up to all facilities providing maternity services? What facilitators do you anticipate?
Acceptability	How well does use of Doppler in intrapartum care align with current national priorities for maternal and newborn care? Are there competing priorities? synergistic factors? Is it likely that using Doppler for intrapartum care will be acceptable to the district medical authorities? facility managers? health care providers providing intrapartum care?

**Table 3 ijerph-17-01931-t003:** Summary of national priorities mentioned by interviewees and alignment to scale-up of Doppler.

Summary Of Responses To: “What Are National Priorities for Improving Intrapartum Care in Tanzania?”	Alignment with Doppler Scale-Up	
	Description	Level *
Ensure competent, high-quality labor and delivery care, including respectful maternity care, to reduce the number of deaths in prenatal and intrapartum care.	Improved fetal heart rate (FHR) monitoring could potentially reduce perinatal death in the intrapartum period.	High
Ensure high-quality newborn care.	Use of Doppler may improve FHR monitoring and thus newborn care.	
During intrapartum care, monitor and document the FHR in the partograph.	Doppler can improve FHR monitoring for better use of partograph.	
Record labor, delivery, and post-delivery client management so the facility can review care.	Doppler can help improve FHR information for quality of care review or for perinatal death audit.	
Increase the number of skilled birth attendants; ensure sufficient supply of lifesaving commodities, equipment, and medicines; and build health care provider capacity.	Having a sufficient supply of Doppler devices may help save newborn lives.	Medium
Manage preterm babies in regional hospitals.	Use of Doppler for FHR monitoring may save lives of preterm babies for treatment in newborn intensive care units.	
Ensure that every mother delivers at a facility with a skilled provider.	Clients may prefer Doppler, which may contribute to better experience and thus higher attendance at care.	Low
Upgrade facilities to provide comprehensive emergency obstetric and newborn care (CEmONC), i.e., cesarean services; promote facility deliveries, early booking, and regular antenatal care (ANC) attendance.	Use of Doppler to monitor FHR may result in more referral to cesarean services. However, use of Doppler is not necessarily associated with the upgrade from BEmONC to CEmONC.	
Book ANC appointments early.	Use of Doppler for intrapartum FHR monitoring is not connected to ANC.	None
Build new health facilities.	Use of Doppler for intrapartum FHR monitoring is not connected to building new health facilities	

* “Levels” (high, medium, low, none) were assigned by authors, not participants.

**Table 4 ijerph-17-01931-t004:** Policy and organizational-level factors affecting scale-up of Doppler in Tanzania.

Level	Key Finding/Needs for Scale-Up	Lessons Learned from HBB Scale-Up
Policy-enabling environment (Government of Tanzania and international donors)	Develop guidelines for health care providers to improve case management upon detection of abnormal FHR in intrapartum care. Fund purchase of Dopplers/training for health care providers. Provide national monitoring system to track results of scale-up of Doppler. Provide guidance on supervision to subnational-level government (supervision or quality checklists).	Use both costing and program monitoring data to track program results.
Nonstate actors/organizational environment (national and international civil society organizations and professsional associations)	Support rollout of Doppler through training and quality assurance activities. Support documentation of use of Doppler in facilities (challenges and benefits).	Training approaches should be evidence-based, and include onsite, low-dose, high-frequency training and clinical mentoring.
Subnational level (regions and districts)	Scale-up should be accompanied with monitoring system. Onsite training should be used; low-dose, high-frequency training preferred; provide clinical mentorship following training.	Sufficient supervisory and technical skills must be available at the district level.

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
