# Peer review of "Scale-Up of Doppler to Improve Intrapartum Fetal Heart Rate Monitoring in Tanzania: A Qualitative Assessment of National and Regional/District Level Implementation Factors"

_ijerph, 2020, doi:10.3390/ijerph17061931_

Round 1

Reviewer 1 Report

Thank you for inviting me to review this interesting paper. I have minor comments to make

As a maternal fetal medicine specialist I was uncertain reading the abstract what exactly you meant by Doppler - hand held Doppler used in antenatal care and in labour or Doppler ultrasound of the fetal vessels? This could be clearer in the abstract Last paragraph in the introduction is perhaps more suitable for the methods in its detail The figures cannot be read as stand alone - suggest explain Please give an example of what you mean by "low dose high frequency" education (it reads pretty explanatory but examples help to clarify!) I was interested that participants were worried about a power source for the Doppler. This may be lost in the cultural context, but I normally use hand held or manually powered dopplers - do you mean using an ultrasound machine to check the fetal heart? In this context I wonder if the participants understand what is meant by Doppler?

Author Response

Reviewer 1

Comment 1: As a maternal fetal medicine specialist I was uncertain reading the abstract what exactly you meant by Doppler - hand held Doppler used in antenatal care and in labour or Doppler ultrasound of the fetal vessels? This could be clearer in the abstract

We thank Reviewer 1 for pointing this out. We have revised the Abstract to be clearer in this regard. We made sure to detail the function of the Doppler, which is as an alternative to the Pinard stethoscope, which is the current standard of care for intermittent fetal heart rate monitoring in low-resource settings. Here is the added sentence:

Handheld Doppler devices have been investigated in several low-resource countries as an alternative to Pinard stethoscope and are more sensitive to detecting accelerations and decelerations of the fetal heart as compared to Pinard.

Comment 2: Last paragraph in the introduction is perhaps more suitable for the methods in its detail.

Thank you for that suggestion, we have moved some of the paragraph to Methods section, but prefer to make the research priorities clear in the Introduction.

Comment 3: The figures cannot be read as stand alone - suggest explain.

Thank you to the Reviewer for letting us know. We have revisited both of the figures in the paper and made the following adjustments:

Figure 1: reworded the title of the figure to be more specific. Included more information in the text surrounding the Figure. Figure 2: provided clarifications on acronyms

Comment 4: Please give an example of what you mean by "low dose high frequency" education (it reads pretty explanatory but examples help to clarify!)

Thank you for this suggestion. We have incorporated two examples with references in the Discussion and hope that this will allow the reader to follow up with more in-depth literature explaining the approach. Therefore, we are not making any revisions to the text.

A recent study in Mozambique found that LDHF training on newborn resuscitation helped improve midwife performance [41], and in Tanzania a supportive intervention to HBB resulted in higher retention of newborn resuscitation skills [23]. A cluster randomized trial in Ghana found that in health facilities using an onsite, LDHF approach to training on intrapartum care, the relative risk of newborn mortality and intrapartum stillbirth significantly declined up to a year following training, leading the authors to recommend use of this training approach [42]

Comment 5: I was interested that participants were worried about a power source for the Doppler. This may be lost in the cultural context, but I normally use hand held or manually powered dopplers - do you mean using an ultrasound machine to check the fetal heart? In this context I wonder if the participants understand what is meant by Doppler?

Thank you for this comment. The concern about lack of power is related to not being able to recharge the device. We do believe that the respondents understood what is meant by Doppler for intermittent fetal heart rate monitoring and in their responses they were noting a concern for being able to recharge the device. The arising suggestion included consideration of manually powered Doppler.

We made an adjustment to reflect this. The sentence on page 8 now reads:

Multiple interviewees mentioned lack of a reliable power source at health facilities to recharge the battery for the hand-held device as a potential barrier to scale-up.

Reviewer 2 Report

The authors declared in the abstract that “this study assessed the perspectives of high-level policymakers on facilitators and barriers to scaling up Doppler for assessing and monitoring intrapartum fetal heart rate (FHR) in Tanzania…”

It is understandable from the method described that authors have interviewed only six officials and two “subject matter experts” those are in Tanjania and another official in the US and assessed their perception on scaling-up of Doppler replacing stethoscope, however, there is no data for real implementation or future implementation. So, they need to modify the title to reflect the fact.

Conclusion section doesn’t seem to reflect the findings of their discussion.

Author Response

Reviewer 2

Comment 1: The authors declared in the abstract that “this study assessed the perspectives of high-level policymakers on facilitators and barriers to scaling up Doppler for assessing and monitoring intrapartum fetal heart rate (FHR) in Tanzania…”

It is understandable from the method described that authors have interviewed only six officials and two “subject matter experts” those are in Tanjania and another official in the US and assessed their perception on scaling-up of Doppler replacing stethoscope, however, there is no data for real implementation or future implementation. So, they need to modify the title to reflect the fact.

Thank you to the reviewer for comments. We have clarified in the manuscript that all experts interviewed were Tanzanian, adding an additional sentence onto page 5 and clarifying in Table 1 that the USAID expert interviewed was from the Tanzanian office, and was in fact Tanzanian.

We also thank the reviewer for providing opportunity to review the title. After careful review, we have decided to slightly modify the title. The old title reads:

Scale-up of Doppler to improve intrapartum fetal heart rate monitoring in Tanzania: a qualitative assessment of national and subnational implementation factors

And the new title reads:

Scale-up of Doppler to improve intrapartum fetal heart rate monitoring in Tanzania: a qualitative assessment of national and regional/district level implementation factors

The new title is more specific as to exactly which level the assessment addressed, as subnational could read as “health facility level” which was not addressed by the research questions. In regards to the title’s mention of implementation factors, we have decided to keep the phrasing as is. In reviewing the research questions:

What are the facilitators and barriers to scale-up of Doppler? What lessons can be learned from Tanzania’s experience of scale-up of the HBB program? Who needs to do what to scale up Doppler in Tanzania?

The authors feel that it is reasonable to describe these as implementation factors. Thus we have elected to keep the title as is, with the slight revision to clarify “subnational.”

Comment 2: Conclusion section doesn’t seem to reflect the findings of their discussion.

Thanks to the reviewer for that observation. We reviewed the Conclusion and cross-checked that all statements are reflective of the Discussion. We also shortened and rephrased for clarity. The revised Discussion reads:

Interviewed experts and policymakers consider scale-up of Doppler for improving intrapartum FHR monitoring as aligning with national health priorities in Tanzania. Linkages and lessons learned from the scale-up of the national HBB program include implementing a structured LDHF training approach that includes clinical mentoring and monitoring of results. It will be useful for the GoT to assess evidence on benefits of Doppler for FHR monitoring using WHO guidance for synthesis of policy, and if Doppler is adopted at scale, to create a scale-up framework from early phases to enhance coordination. If scale-up is pursued, sufficient resources, training and infrastructure to support effective use of Doppler for FHR monitoring and intrapartum case management must be in place to overcome barriers described by participants.

Round 2

Reviewer 2 Report

The authors really could not be effectively improved their manuscript. They changed some sentences but that does not address the prior concern mentioned. The authors basically submitted the same manuscript with a few sentences changed. The importance of the work is not reflected in the manuscript. The major weakness is its conclusion part. They couldn't be able to establish why their manuscript would be important to scientific or any other community.
